# Study of Interaction Mechanism between Positrons and Ag Clusters in Dilute Al–Ag Alloys at Low Temperature

**DOI:** 10.3390/ma14061451

**Published:** 2021-03-16

**Authors:** Xiaoshuang Liu, Peng Zhang, Baoyi Wang, Xingzhong Cao, Shuoxue Jin, Runsheng Yu

**Affiliations:** 1Institute of High Energy Physics, Chinese Academy of Sciences, Beijing 100049, China; liuxs@ihep.ac.cn (X.L.); zhangpeng@ihep.ac.cn (P.Z.); wangboy@ihep.ac.cn (B.W.); caoxzh@ihep.ac.cn (X.C.); jinshuoxue@ihep.ac.cn (S.J.); 2School of Nuclear Science and Technology, Chinese Academy of Sciences, Beijing 100039, China

**Keywords:** Al–Ag alloys, Ag clusters, positron annihilation, monovacancies, shallow trapping

## Abstract

The microstructural evolution of dilute Al–Ag alloys in its early aging stage and at low temperatures ranging from 15 K to 300 K was studied by the combined use of Positron annihilation lifetime spectroscopy (PALS), high resolution transmission electron microscopy (HRTEM), and positron annihilation Coincidence Doppler broadening (CDB) techniques. It is shown that at low temperatures below 200 K, an Ag–vacancy complex is formed in the quenched alloy, and above 200 K, it decomposes into Ag clusters and monovacancies. Experimental and calculation results indicate that Ag clusters in Al–Ag alloys can act as shallow trapping sites, and the positron trapping rate is considerably enhanced by a decreasing measurement temperature.

## 1. Introduction

Al–Ag alloys are important for the study of the fundamental processes of decomposition in supersaturated alloys, owing to the large difference in the atomic numbers of Al and Ag (13 versus 47) [1]. Although Al–Ag alloys have limited structural applications, Ag is a commonly used microalloying element in aluminum alloys. Adding a small amount (from 0.1 at.% to 0.5 at.%) of Ag into aluminum-based alloys can improve their mechanical properties, thermal stability, and stress corrosion cracking resistance [2,3]. This effect is widely seen in aluminum alloys, particularly in the high-strength alloys for advanced aerospace and defense applications, including Al–Cu–Mg and Al–Zn–Mg-based alloys [4,5,6]. However, most studies on Al–Ag alloys have focused on the relatively high Ag concentration in Al–Ag alloys. In 1942, Guinier [7] found that the Ag atoms in Al–Ag alloys first form small clusters, then evolve into Guinier-Perston (GP) zones and further larger precipitates in the early stage of the aging process. Guinier-Preston (GP) zones are the early-stage solute enriched regions in aluminum alloys [8,9]. Nagai et al. [10] claimed that the chemical composition of Ag-rich nanoclusters is AlAg_3_ in aged Al-2.0 at.% Ag from experimental and calculated two-dimensional angular correlation of positron annihilation radiation (2D-ACAR) anisotropy results. In a recent study, B. Zou et al. [11] reported the precipitation process of Ag nanoclusters in three Al–Ag alloys with different Ag contents (Al-1 wt.%Ag, Al-5 wt.%Ag, and Al-15 wt.%Ag). However, in the above studies, little attention was paid to the Al–Ag alloys with a very low Ag concentration (less than 0.5 wt.%). Moreover, the mechanism of formation and migration of Ag clusters in Al–Ag alloys at the early stage of aging is still unclear. The main reason for this is most probably that the Ag clusters that form at the early stage of aging in dilute Al–Ag alloys are too small to be detected by conventional methods. In particular, unlike most other aluminum alloys, GP zones in the Al–Ag system form immediately after quenching [12,13], which makes it difficult to detect the precipitation of Ag clusters in the early stage of aging.

Positron annihilation spectroscopy (PAS) is a highly sensitive technique for studying subatomic scale defects, e.g., 10^−5^~10^−6^ density defects, in metals and semiconductors [14,15,16]. If a vacancy exists in the material, it will easily capture positrons, since the vacancy does not contain positively charged nuclei. In fact, there are two typical sites in which positrons can be trapped [17,18]. One is the deep positron trapping sites, such as vacancies and voids—since the positron affinity of these defects is less than −1 eV [19], positrons encounter a deep potential well from which it is thereby difficult to escape; the other type is the shallow positron trapping sites—e.g., dislocations. The positron affinities of these defects are greater than −100 meV [20]. Hidalgo et al. [21] studied positron annihilation in deformed copper and found that dislocations can also trap positrons, and the positron trapping rate increased when the temperature was below 77 K. In addition to the two above-mentioned typical types of positron trapping sites, in recent years, some clusters have been shown to be able to act as shallow positron trapping sites; e.g., in 2002, Huis et.al [22] found that lithium nanoclusters are able to trap positrons in MgO, and the trapping ability is enhanced at low temperatures. Indeed, the reason for positron trapping lies mainly in the different chemical elements. A table of positron affinities for most elements can be found in the work of Puska et al. [23]. The values are −4.41 and −5.36 eV for aluminum and silver, respectively. This indicated that the Ag atom has a weak but extended attractive potential to positrons in Al–Ag alloys. Zou et al. [11] found that the Ag clusters in Al–Ag alloys can act as positron trapping centers and that the confinement of positrons is enhanced with the increase in the size of Ag clusters. Moreover, Hugenschmidt et al. [24] found that coincidence doppler broadening (CDB) is sensitive to the detection of small precipitates in aluminum, and even a 0.1 nm thin Sn layer embedded underneath a 200 nm Al coating leads to a significant and unambiguous fingerprint in the CDB spectrum. Nevertheless, studies on the positron trapping mechanism in dilute Al–Ag alloys have not been reported in literature. Moreover, it is very important to investigate the formation and evolution of Ag clusters in Al–Ag alloys.

The main purpose of this work is to investigate the microstructural evolution of Al–Ag alloys in the early aging stage by means of positron annihilation spectroscopy to examine whether or not the minor Ag clusters can act as positron trapping centers and to study the temperature dependence of positron annihilation behavior. In addition to positron annihilation lifetime spectroscopy (PALS) and coincidence doppler broadening (CDB) measurements, we also used high resolution transmission electron microscopy (HRTEM) to investigate the Ag clusters at the early aging stage in Al–Ag alloys.

## 2. Materials and Methods

### 2.1. Materials and Preparation

The Al-0.2 wt.%Ag alloys were prepared from aluminum (99.999% purity) and silver (99.999% purity) using a high-frequency induction furnace in a vacuum environment. After melting, the specimens were solution treated at 823 K for 24 h in a vacuum environment, quenched by ice water as quickly as possible (~15 ms), thinned to 1 mm, and punched into 10 × 10 mm^2^ square sheets. Pure aluminum and sliver samples were also prepared as reference specimens. All the square sheet samples were electrochemically polished to a mirror-like surface using 25% HClO_4_ (Beijing Institute of Chemical Reagents, Beijing, China) and 75% C_2_H_6_O (Beijing Institute of Chemical Reagents, Beijing, China) polishing solution at room temperature (300 K).

Al-0.2 wt.%Ag alloys were then heated at 823 K for 30 min, followed by quenching into ice water. Then, the specimens were naturally aged for 24 h at room temperature. The specimens were measured PLAS and DBS at temperatures of 15–300 K for about 12 h. After that, the CDB spectra were measured for about 12 h.

### 2.2. Characterization Methods

PALS measurements were carried out using a fast–slow coincidence system with a time resolution of 196 ps (full width at a half maximum). A ^22^Na positron source with an intensity of 10 μCi was encapsulated by a Kapton foil (made by Nilaco Corporation, Tokyo, Japan). The thickness of the Kapton foil was 7.5 microns. The total number counts were 2.0 × 10^6^ for each lifetime spectrum. The count rate of the PALS was about 100 cps. The data from the experiment were analyzed using the LT9 program [25]. After deducting the components (12%) of the positrons annihilated in Kapton foils, one or two lifetime components were identified in our experimental spectra. At 15–200 K, the positron annihilation lifetime could only be deconvoluted into one lifetime component. Above 200 K, the positron annihilation lifetime was decomposed into two components.

Doppler broadening spectroscopy (DBS) was performed using a single high-purity Ge detector (GEM35P4-76-PL, ORTEC, Oak Ridge, TN, United States) with an energy resolution of 1.3 keV at 511.0 keV. The total number of counts for each single detector DBS spectrum was also 2.0 × 10^6^. The *S* and *W* parameters were obtained from these DBS results. The *S* parameter was defined as the ratio area in 510.55–511.45 keV to the total area of the 511 keV photo peak, reflecting the ratio of the annihilation of positrons with the low-momentum electrons. The *W* parameter is defined as the ratio area in 514.6–521.2 keV and 500.8–507.4 keV to the total area, reflecting the ratio of annihilation of positrons with the high-momentum electrons. CDB measurements were carried out using two high-purity Ge detectors, with the efficiency of 35%. The counting rate was about 100 cps, and each spectrum took 12 h to ensure no less than 10^6^ counts in each CDB spectrum. To obtain supplementary information about the Al–Ag sample microstructure, a TecnaiF30 high resolution transmission electron microscopy (HRTEM, FEI, Hillsboro, OR, United States) operating at 300 kV was utilized.

## 3. Results and Discussion

Figure 1 shows a typical HRTEM image of Al–Ag alloy sample after aging at room temperature for 24 h. The inset in Figure 1 shows the Fast Fourier Transform (FFT) of the HRTEM image. The HRTEM presented almost prefect crystallographic image. There were no defects or clusters that could be observed in the HRTEM image. The reason may be that the atomic size difference between Al (1.43 Å) and Ag (1.44 Å) is negligible. Thus, it was difficult to distinguish clusters using HRTEM. Moreover, the concentration of Ag clusters in Al-0.2 wt.%Ag alloys is too low (less than 2 ×10^−4^). The HRTEM was not sufficiently sensitive to detect such a low content of clusters. It is thus understandable that most previous studies focused on samples with a high Ag content after aging at high temperature [26], where the sizes of GP zones in the studied samples were more than 10 nm. However, as proved by the present study, it is difficult to detect Ag cluster in Al–Ag alloys with very low Ag content after aging at room temperature, even with HRTEM. The FFT of the HRTEM image however has streaks which may indicates the presence of some very fine linear defects or clusters. To make further clarification, positron annihilation spectroscopy is thereafter applied for characterization of the nanostructure in dilute Al–Ag alloys.

The positron annihilation lifetime of quenched Al-0.2% Ag alloys at different temperatures ranging from 15 K to room temperature was obtained by LT9 analysis. At 15–200 K, the positron annihilation lifetime could only be deconvoluted into one lifetime component *τ_m_* about 150 ps after deducting the source components. We measured the positron annihilation value of pure Al at room temperature. The observed value was 150 ps, which was very close to that in the Ag–Vacancy complex (146 ps) [27]. In addition, the positron annihilation lifetime in the vacancy is 231 ps [28], which is much larger than the positron annihilation lifetime at 15–200 K. It is worth noting that the positron annihilation lifetime at 15–200 K is very close to that in aluminum and Ag–V complexes. This provides evidence that the V–Ag complexes were formed after being quenched, which is consistent with the conclusion of Nagai et al. [27] who showed that quenched-in vacancies are bound to Ag atoms in Al–Cu–Mg–Ag alloys with very low Ag contents. Furthermore, Shasha Zhang et al. [29] reported that the Ag atoms have a strong attraction interaction with the vacancy in Al–Ag alloys. That may be the origin of the formation of Ag–V complexes in Al–Ag alloys. In 1988, F. Maury et al. [30] conducted electrical resistivity measurements on dilute Al–Ag alloys after electron irradiation. They reported that the Ag atoms in the dilute Al–Ag alloys are easily trapped by vacancies to form Ag–V complexes when the temperature is below 180 K, and Ag–V complexes can be recovered above 180 K. This coincides rather well with our current positron annihilation measurement results.

As shown in Figure 2, the positron annihilation lifetimes in Al–Ag alloys changed from one component of ~150 ps to two components of 118 ps and 195 ps over 200 K as the temperature increased. Most probably, in our studied dilute Al–Ag alloys, Ag–V complex formed as they were quenched, and after 200 K, the Ag–V complex decomposed into Ag clusters and vacancies. With Ag atoms gradually separated from the binding of the vacancies, the size of the open space near Ag–V complexes increased, resulting in the appearance of a long positron annihilation lifetime of 195 ps (*τ*_2_). Indeed, above 200 K, the short lifetime (*τ*_1_) of around 118 ps was close to the positron annihilation lifetime of Ag clusters, and the long lifetime of 195 ps was close to the positron annihilation lifetime of monovacancies [31]. This provides evidence that the Ag–V complexes are decomposed into monovacancies and Ag clusters at 200 K. Thereafter, part of the injected positrons annihilated in the vacancies, while others annihilated on Ag clusters. Furthermore, as can be seen from Figure 2, the relative intensity (*I*_1_ and *I*_2_) for *τ*_1_ and *τ*_2_ gradually decreased from 52.7 to 44 and increased from 47.3% to 56%, respectively, as the temperature increased from 200 K to 300 K. That indicates that fewer positrons annihilated with the electrons around Ag clusters and more positrons annihilated with the monovacancies as the temperature increased. Thus, the ability of Ag clusters to capture positrons is enhanced at a lower temperature.

Single detector DBS results are shown in Figure 3. It is evident that the *S* increased from 0.494 to 0.509 as the temperature increased from 15 K to 200 K. The *W* showed an opposite trend, decreasing from 0.112 to 0.097. This suggests the same conclusion as that of the PALS results; i.e., Ag atoms gradually separated from the Ag–V complex and resulted in an increase in the size of the open space near Ag–V complexes at 200–300 K. It is worth noting that the *S* value started to increase slowly after 200 K. In particular, from 250 K, the *S* value basically remained stable. This indicates that almost all Ag–V complexes decomposed into Ag clusters and monovacancies until 250 K.

Figure 4 shows the DBS results with the *S-W* plot of Al–Ag alloys at different temperatures. Clearly, the *S-W* characteristic points are in a straight line, and the *S-W* characteristic points move from Ag towards Al with increasing temperature. This indicates that the surrounding elements in which positrons were trapped and annihilated gradually changed from Ag clusters (surrounding Ag–V complexes) to aluminum (surrounding monovacancies) as the temperature increased. This is in consistent with the above-mentioned conclusion from the positron annihilation lifetime results. 

Figure 5 shows the ratio curves for CDB spectra of Al-0.2% Ag at different temperatures compared to pure Al; the ratio curve for pure Ag measured at room temperature is also added for comparison. A CDB ratio curve can reflect the momentum distribution of core electrons of certain elements annihilated with positrons, due to the significantly enhanced signal-to-noise ratio of at least 10^5^:1 for a typical CDB spectrum. As shown in Figure 5, there is a narrow peak at around 11 × 10^−3^
*m*_0_*c* in the CDB ratio curve of pure Ag, which is consistent with the experimental results of Y. Nagai [32] and B. Zou [11]. The curves in Figure 5 show a bump at 11 × 10^−3^
*m*_0_*c* at low temperatures, which disappears at high temperatures. In particular, at 200 K, the characteristic peak at 14 × 10^−3^
*m*_0_*c* moves to 15 × 10^−3^
*m*_0_*c*. Combined with the positron annihilation lifetime results, this gives evidence that Ag atoms gradually become free from binding with a vacancy and the Ag–V complexes decomposed to Ag clusters and monovacancies at 200 K. When the temperature reached 300 K, however, the characteristic peak for Ag clusters in Al-0.2% Ag was hardly observable; there was only a very small ratio fluctuation around 1.0.

When the temperature increased above 200 K, the value of characteristic peak gradually decreased with the increase in temperature. That is, above 200 K, more positrons annihilated with low-momentum electrons around monovacancies and fewer positrons were trapped and annihilated with high-momentum inner shell 4d electrons of Ag clusters. In other words, the Ag clusters can trap positrons, but the positron trapping ability of Ag clusters decreases with the increase in temperature. At room temperature, the Ag clusters could barely trap positrons any more. Based on such temperature-dependent behavior, we may deduce that the Ag clusters can act as shallow positron trapping sites to trap positrons, and the trapping ability increases with the decrease in temperature. The conclusion is consistent with the results of B. Zou et al. [11].

Puska et al. [23] have reported that a minimum radius of embedded clusters is necessary for positron trapping to occur. There is one bound state if the radius of the precipitate is larger than the critical radius, $r_{c}=\frac{5.80a_{0}}{\sqrt{\Delta A}}$, where *ΔA* is the positron affinity difference between the two kinds of metals, and *a*_0_ is the Bohr radius. For the Al–Ag alloy, the critical radius is about 5.9 *a*_0_ with *A**_+_* (Al) = −4.41 eV and *A**_+_* (Ag) = −5.36 eV. This means that precipitates should contain at least ~6 Ag atoms.

We attempted to quantitatively describe the positron trapping mechanism in Al–Ag alloys at varied temperatures utilizing the three state trapping model [20,33]. At 15–200 K, there were two positron trapping states in the Al–Ag alloy: free annihilation (bulk state) and deep trapping (Ag–V complex). At 200–300 K, there were three positron states in the Al–Ag alloy: free annihilation (bulk state), shallow trapping (Ag clusters), and deep trapping (vacancies). For spherical defects, the positron trapping rate and detrapping rate capture rate can be written as [19,34]
(1)$$\kappa \left(T\right)=r_{0}Ce^{-\gamma T}$$
(2)$$\delta \left(T\right)=\frac{m}{4\hbar^{2}}\pi^{-\frac{1}{2}}C^{-1}\kappa \left(T\right){\left(k_{B}T\right)}^{\frac{3}{2}}E_{b}^{-\frac{1}{2}}e^{-\frac{E_{b}}{k_{B}T}}$$
where *γ* is the trapping rate of positrons by the trapping sites at 0 K and *r*_0_ is a constant representing the specific trapping rate of such trapping sites. *C* means the concentration of trapping sites, *E_b_* means the binding energy of the trapping sites and positrons. The above equations show that the trapping rate is proportional to the concentration of trapping sites (either shallow or deep trapping). In contrast, the detrapping rate is independent of the trapping site concentration but related to the binding energy.

Using the above equations, we calculated the temperature dependence of the positron detrapping rate as a function of temperature for a range of binding energies from 0.01 eV to 1 eV. The calculated results are shown in Figure 6. It can be seen that the higher the binding energy, the smaller the detrapping rate. When the binding energy reaches 1 eV, the detrapping rate remains at around 0. This suggests that the higher the binding energies, the harder it is for the positrons to escape from the trapping sites. As regards the temperature dependence, it is clear that an increased temperature facilities the detrapping of positrons from trapping sites for all the binding energies except *E_b_* = 1 eV. It was reported that the positron binding energy for Ag clusters is lower than 40 meV [11]; therefore, Ag clusters could only act as shallow trapping defects for positrons, and the low temperature enhancement of the effect of Ag on positron trapping is thereby understandable.

## 4. Conclusions

In summary, we conducted a positron annihilation spectroscopy study on the early stage of precipitation in dilute Al–Ag alloys. The interaction mechanism between positrons and Ag clusters as well as monovacancies in Al–Ag alloys at varying temperatures were extensively studied. Several conclusions can be drawn. Firstly, a monovacancy in quenched Al–Ag alloys can easily combine with Ag atoms to form Ag–V complexes at 15–200 K, and within this temperature range, the Ag–V complex effect is predominant for positron annihilation in the quenched Al-0.2% Ag alloy. Secondly, above 200 K, Ag atoms gradually escape from binding with vacancies and migrate to form Ag clusters. Meanwhile, Ag clusters can act as shallow trapping defects, and the positron detrapping from Ag clusters is enhanced at higher temperatures. Last but not least, the addition of a small amount (0.2 wt.%) of Ag has a great influence on the positron trapping mechanism; in other words, low-temperature positron annihilation spectroscopy is an important method to investigate the microstructural evolution of defects and precipitates in the early aging stage of dilute alloys.

## Figures and Tables

**Figure 1 materials-14-01451-f001:**
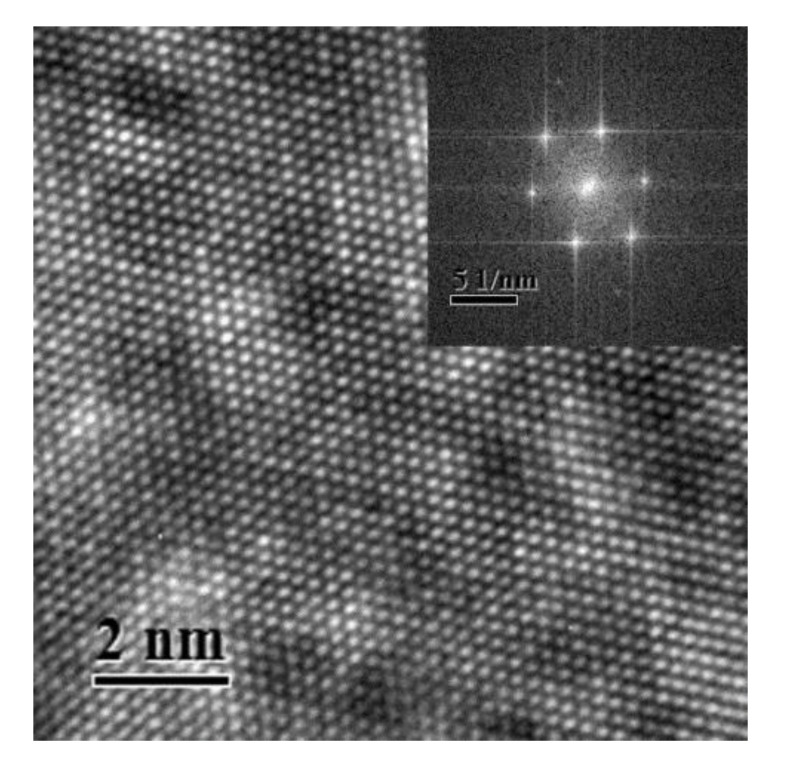
High resolution transmission electron microscopy (HRTEM) image of the nanostructure of Al-0.2 wt.%Ag sample aged at room temperature after quenching. The inset view is the Fast Fourier Transform (FFT) of the HRTEM image.

**Figure 2 materials-14-01451-f002:**
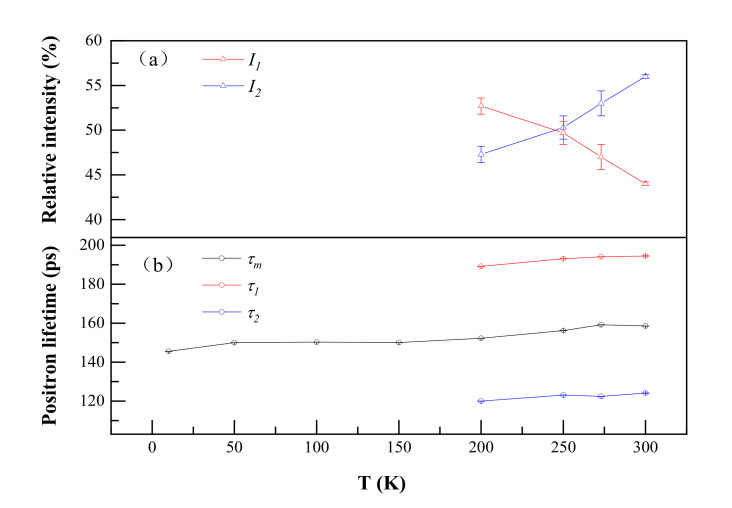
Positron annihilation lifetimes (**a**) and their relative intensities (**b**) of Al-0.2 wt.%Ag as a function of temperature. *τ*_1_, *τ*_2_ and *τ_m_* represent a short lifetime, long lifetime and mean lifetime in the positron annihilation spectrum. I_1_ and I_2_ represent the relative intensity of the short lifetime and long lifetime.

**Figure 3 materials-14-01451-f003:**
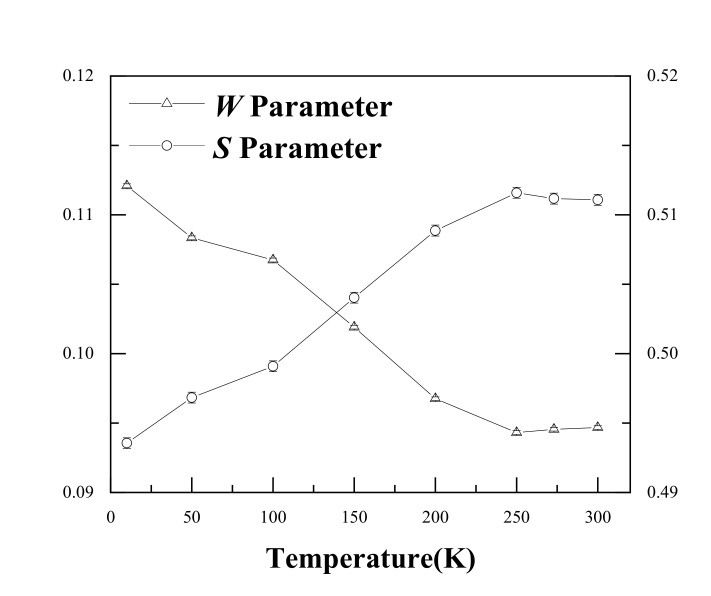
*S* and *W* values for the Al-0.2% Ag alloys at different measurement temperatures.

**Figure 4 materials-14-01451-f004:**
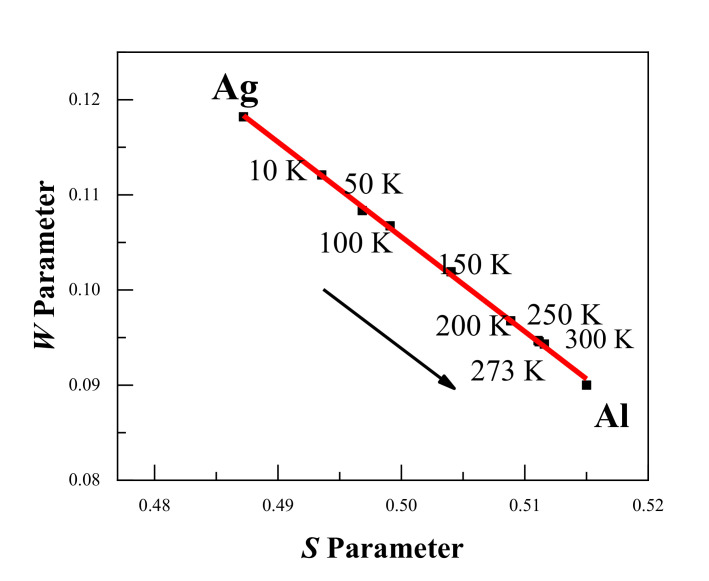
*S-W* plot for the Al-0.2 wt.%Ag alloys at different temperatures. The arrow shows the increase in the measurement temperature, and the movement of the *S-W* characteristic points from Ag towards Al. The red line is presented as a visual aid.

**Figure 5 materials-14-01451-f005:**
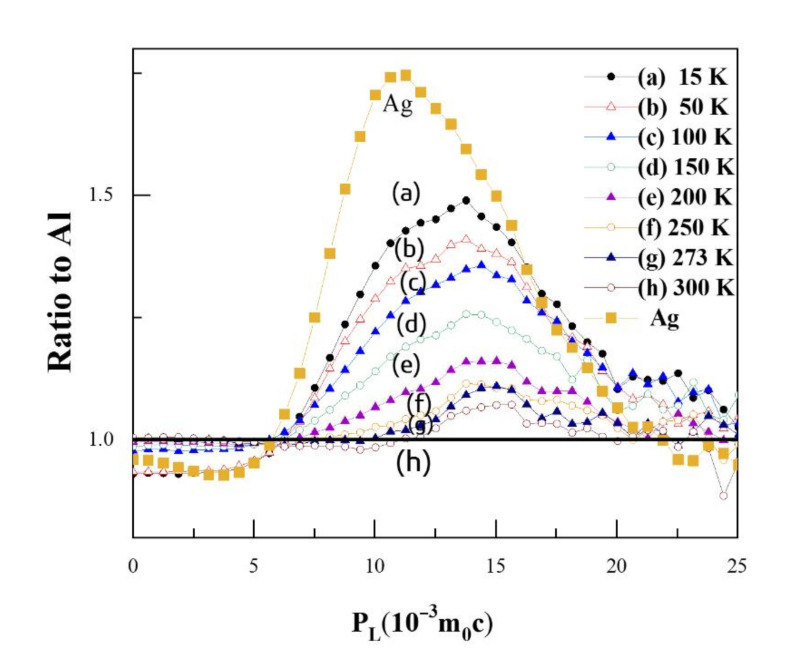
Coincidence doppler broadening (CDB) ratio curves for Al-0.2 wt.%Ag at different temperatures. The statistical errors are omitted for clarity of representation.

**Figure 6 materials-14-01451-f006:**
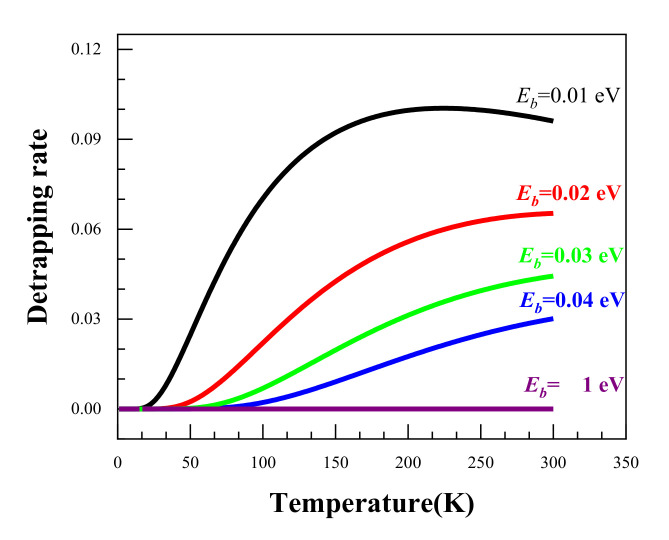
The temperature dependence of the positron detrapping rate at different binding energies.

## Data Availability

Data sharing not applicable, all the data created for this study are already displayed in the article.

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
