# Peer review of "Study of Interaction Mechanism between Positrons and Ag Clusters in Dilute Al–Ag Alloys at Low Temperature"

_materials, 2021, doi:10.3390/ma14061451_

Round 1
Reviewer 1 Report
The submitted manuscript discusses the interaction mechanism between positrons and Ag clusters in dilute AlAg alloys at low temperature. The topic is of interest for several applications in the fields of electronic technology, automotive body structure, wind and solar energy management. The authors used experimental methods to verify the Ag cluster affinity to trap positrons. The English language requires improvement in the entire manuscript. The references style is not well followed and literature data is not sufficient to tackle the topic idea. It was also noticed that the microstructural analysis was not adequate to explain the clustering phenomenon in AlAg alloy. Detailed comments can be found in the attached file.

Reviewer 2 Report
The work discusses interesting and important results on diluted Al-Ag alloys by means of PALS and CDBS measurements. The methods are properly chosen. The interpretation of the results needs some extra work.
The work deserves to be published after a major revision.
My comments are introduced in the attached PDF file.

Reviewer 3 Report
The paper by R. Yu et al. reports an experimental study on the early stage of precipitation in AlAg alloys characterized by a small concentration of silver. The paper is generally well written, the results are nice and the discussion is exhaustive. I propose to publish the paper after the authors have considered the following minor remarks:
- line 88: use of Kapton as support of the positron source should be explained, otherwise a reader not acquainted with PAS techniques could be disappointed.
- figure 2, caption: ‘variance’ can be ambiguous, due to its statistical meaning. Maybe it could be substituted with ‘variation’, or ‘behaviour’.
- line 137: ‘long positron annihilation’: the word ‘lifetime’ should be added.
- line 137: there is an ‘and’ which should be erased, unless the authors missed some words.
- line 204: ‘agreeable’ should be changed into in ‘agreement’
Some typos:
- line 34: ‘how the Ag atoms accumulation…’ : maybe ‘accumulate’?
- line 88: positrons ‘annihilated’
Round 2
Reviewer 1 Report
Thanks to the authors for responding adequately to the reviewer's comments.
Reviewer 2 Report
I am completely satisfied with the introduced changes in the paper revision. Still the authors missed putting some physical quantities and constants into italics (for example mo, c, Eb) but this can be corrected in the checking proofs process.
